# Synthetic Realism in Finance: A High-Fidelity Market Simulation with Adaptive LLM Agents

## Abstract

Agent-based modeling offers a powerful framework for understanding complex systems, from social interactions to ecological networks. In finance, such models are crucial for stress testing, risk assessment, and the development of intelligent trading strategies. However, the fidelity of these simulations is often limited by the simplicity of their agents, which typically operate on fixed, rule-based heuristics. Here we present a novel, high-fidelity financial market simulation environment that integrates Large Language Models (LLMs) to create adaptive, reasoning-based agents. We augment the well-established ABIDES simulator with a novel Brain-Memory Architecture. This features a core LLM "Brain" for complex, real-time decision-making, supported by a persistent memory module that records the outcomes of its past actions. This enables a powerful online learning loop, allowing the agent to adapt its strategy based on immediate feedback within a single simulation run. By ingesting real-world NASDAQ data at the message level and introducing a calibrated population of background traders, we demonstrate that our LLM-enhanced system accurately reproduces key market microstructure phenomena, including price discovery and trade intensity. This work establishes a new paradigm for synthetic financial environments, providing an unprecedented platform to study the emergent dynamics of human-like intelligence in complex, competitive systems and enabling the development of next-generation, explainable AI agents for financial applications.

## 1   Introduction

The modern financial landscape is a complex adaptive system, characterized by intricate, non-linear dynamics that are difficult to model with traditional econometric methods [3, 11]. The need for high-fidelity synthetic data has become paramount for developing robust trading strategies, stress-testing financial systems, and training sophisticated artificial intelligence (AI) agents [19, 10]. While real-world market data is a cornerstone of financial research, its inherent privacy concerns, inaccessibility, and often incomplete nature limit its utility for large-scale, controlled experimentation [19].

Synthetic data, artificially generated to mimic the statistical properties and patterns of real-world data, is emerging as a powerful solution to these challenges. In a highly regulated industry like finance, it provides a means to address critical issues of privacy, fairness, and explainability. It allows financial institutions to train and validate AI models without compromising sensitive information, to create balanced datasets that mitigate algorithmic bias, and to model rare but high-impact events that are underrepresented in historical records.

Agent-based simulations [5, 2, 6, 15, 14, 17, 20], such as the ABIDES platform, provide a controlled environment for studying market behavior via exchange-style messaging, configurable latency, and limit-order-book mechanics. However, a fundamental limitation remains: most simulations rely

Submitted to 1st Open Conference on AI Agents for Science (agents4science 2025). Do not distribute.

on pre-defined, static strategies (e.g., Zero-Intelligence or Momentum) [9, 12] that cannot process unstructured information, reason about sentiment, or adapt online to shifting regimes.

**Our approach.** We present a high-fidelity market simulator that integrates Large Language Models (LLMs) as adaptive, reasoning agents within an enhanced ABIDES environment. Concretely, we augment ABIDES with (i) a *news-aware LLM layer* that converts structured news events into actionable signals for trading and market-making agents, and (ii) a *Brain–Memory architecture* that supports within-episode adaptation by retrieving prior experiences and writing back outcomes to persistent memory. We replay real NASDAQ TotalView-ITCH data (via LOBSTER) at the message level, calibrate a heterogeneous population of background agents, and validate microstructure realism against historical benchmarks.

At a high level, the EnhancedLLMNewsAnalyzer normalizes LLM analyses of structured news into sentiment, confidence, and risk annotations that directly drive two ABIDES-native agents: ABIDESLLMTradingAgent (direction, strength, duration, and size under position caps) and ABIDESLLMMarketMaker (spread and inventory control that widen/recenter under uncertainty). This coupling yields interpretable dynamics where sentiment drives directional pressure and uncertainty modulates liquidity.

The *Brain–Memory* design separates strategic reasoning from persistent learning: the LLM "Brain" acts on a real-time state vector (market + news) while a file-backed *Memory* retrieves similar past contexts and logs outcomes (parameters, actions, P&L). The loop operates online within a single session: decisions at hour one are evaluated, written back to memory, and inform strategy at hour two.

To ensure *message-level fidelity*, our pipeline parses and replays ITCH events and calibrates agent populations so simulated order flow, volume, and volatility align with historical NASDAQ data. For external validation, we align simulated executions to a per-second ITCH clock and compute price-error in basis points, alongside visual overlays of mid-price and trades.

**Contributions.** This work makes the following contributions:

1. **LLM agents with online adaptation.** A Brain–Memory architecture that retrieves prior experiences and updates memory with realized outcomes, enabling within-episode strategy adaptation.

2. **News-aware intelligence that modulates liquidity and risk.** A structured news interface whose outputs drive an LLM trading agent and an LLM market maker, coupling sentiment to directional pressure and uncertainty to resiliency.

3. **Message-level realism and calibration.** A data pipeline that ingests NASDAQ ITCH (via LOBSTER), calibrates heterogeneous background agents, and validates against historical microstructure using execution-level metrics.

4. **Experimental evidence under matched conditions.** On AMZN (June 21, 2012), we compare LLM-ON, LLM-OFF, and Baseline populations under identical seeds and configurations; LLM-ON perceives injected news and reproduces key microstructure phenomena.

Together, these elements bridge the gap between rule-based simulators and adaptive, explainable agents, yielding a platform for studying emergent behavior in realistic, event-driven markets and for generating synthetic data that captures microstructure-level nuance.

## 2 Related Work

### 2.1 High-Fidelity Agent-Based Market Simulation

Agent-based market simulators enable controlled studies of trading strategies and market dynamics. A prominent example is ABIDES (Agent-Based Interactive Discrete Event Simulation) [5], an open-source framework designed for high-fidelity equity market simulation. ABIDES [5] supports tens of thousands of trading agents interacting with an exchange agent and enforces realistic "market physics", e.g. nanosecond timestamp resolution, configurable network latencies, and standardized exchange messaging protocols. These features allow researchers to replicate continuous double auctions and even replay specific historical trading days with fine-grained control. Built on this foundation, recent works have extended ABIDES for new research purposes. ABIDES-Gym [2] provides an OpenAI

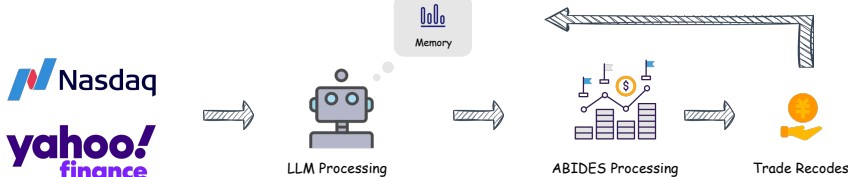

Figure 1: End-to-end pipeline of the LLM-Enhanced ABIDES architecture. Real NASDAQ TotalView–ITCH data (via LOBSTER) are parsed and replayed to drive and calibrate the enhanced ABIDES core (kernel & messaging; limit-order-book and price–time matching; heterogeneous baseline agents). A news-aware LLM layer (EnhancedLLMNewsAnalyzer) converts structured news into sentiment/confidence signals that condition ABIDES agents (ABIDESLLMTradingAgent and ABIDESLLMMarketMaker) operating under exchange constraints. A Brain–Memory module enables within-episode adaptation by retrieving past experiences and writing back outcomes (*news, params, action, P&L*) after evaluation. The simulator emits `OrderBookSnapshot/Trade` streams which are aligned with ITCH to compute microstructure metrics (e.g., price_error_bps) and to render overlay visuals.

Gym interface to the discrete-event simulator, making it easier to train Reinforcement Learning (RL) agents [21] in a multi-agent market environment. For instance, Amrouni et al.[2] wrap the ABIDES-Markets simulator as Gym environments to benchmark daily investor and order-execution tasks, enabling RL agents to interact with a realistic limit order book (LOB) market as their training ground. Likewise, ABIDES-Economist [6] adapts the ABIDES platform to macroeconomics, simulating an entire economy of heterogeneous households, firms, a central bank, and a government. This platform introduce learning-capable agents into this economic simulator, allowing agents to either follow rule-based policies or learn optimal behaviors via interactions in an OpenAI Gym style environment. This extension showcases the versatility of ABIDES beyond financial markets, as it can model complex economic systems with agent learning and exogenous shocks for AI-economics research.

## 2.2 Multi-Agent Reinforcement Learning in Market Simulations

The integration of reinforcement learning [21] with high-fidelity market simulators has been an active research area [17, 20, 15, 14]. Karpe et al. [13] demonstrated a multi-agent RL approach to optimal trade execution using an ABIDES-based LOB simulator. They configured a historical order book scenario in ABIDES and trained a trading agent with Double Deep Q-Learning [16] to execute a large order optimally. Notably, the learned RL agent in some cases converged to a classic Time-Weighted Average Price (TWAP) strategy, a sensible baseline, indicating that the simulator's realism can foster plausibly optimal behaviors. The performance of the RL policy was evaluated against real-market data via market replay, showing the importance of a high-fidelity environment for trustworthy policy testing. This study illustrate how combining ABIDES' realistic market mechanics with multi-agent RL techniques can yield insights into algorithmic trading strategies under near-real conditions.

## 2.3 Simulation Realism and Transferability

A persistent challenge in market simulation research is ensuring that agent behaviors learned in simulation generalize to real markets. Vyetrenko et al. [24] highlight that many simulators [20, 24, 17, 4] "fail to reproduce statistics and stylized facts seen in real markets," undermining the robustness of strategies validated purely in silico. Their work, "Get Real," [24] catalogs a comprehensive set of realism metrics from distributional properties of returns and order flows to higher-order stylized facts to quantitatively compare simulated and real LOB data. The results revealed significant discrepancies between even sophisticated simulations and actual market data, providing a benchmark for improving fidelity. Along similar lines, Pino et al. [18] propose similarity metrics for transfer learning [23] in trading agents to bridge the "sim-to-real" gap [22]. They evaluate conceptual similarity (comparing stylized statistical features of price series) [24, 7], structural similarity (comparing agents' experience trajectories) [1], and performance similarity between different market MDPs [8]. These metrics guide when a policy learned in one market (or a simulator) can be safely transferred to another. By employing techniques like Probabilistic Policy Reuse [8] based on such similarity measures, one can adapt or reuse trading policies from a simulator in the real market context more effectively. This line

of work underlines the importance of quantifying fidelity and alignment between simulated and real environments as a prerequisite for reliable AI trading research.

# 3 LLM-Enhanced ABIDES Architecture

Our system is built on an extended version of the ABIDES simulator, tailored to support and integrate LLM agents. The core innovation lies in a modular architecture that separates the roles of decision-making and memory, leveraging the strengths of different LLMs.

## 3.1 The Limit Order Book and Matching Engine:

At the core of our system is a clean and robust LOB engine, implemented in `order_book.py`. It maintains two distinct sides of the book—bids and asks—as price-indexed FIFO queues and uses a global order map for lifecycle tracking. The engine employs a strict price-time-priority matching algorithm: market orders consume liquidity from the top of the book, while marketable limit orders behave similarly until their limit is reached, with any leftover quantity resting as displayed liquidity. Resting orders preserve their FIFO timestamp at each price level. Every match produces a `Trade` record with detailed information, including timestamp, symbol, price, quantity, aggressor side, and participating order/agent IDs. After each trade, the engine updates key market metrics, including best bid/ask, spread, mid-price (the average of the best bid and ask), and last trade price. It also emits periodic `OrderBookSnapshot` rows, capturing the state of the book and key metrics for subsequent analysis.

## 3.2 ABIDES Agent Population

A Heterogeneous Environment: To create a realistic, multi-agent environment, our system utilizes a diverse population of pre-existing ABIDES agent types as background traders. These agents, which have been validated in prior research, provide a heterogeneous and dynamic market context for our LLM agents. The key agent types include:

- `Zero-Intelligence (ZI) Agents:` These agents, governed by simple stochastic rules, provide a fundamental layer of random, unpredictable behavior, mimicking market "noise."
- `Fundamental Value Agents:` These traders are informed by real or simulated historical price data, placing orders based on their belief in a stock's intrinsic value.
- `Market Study Agents:` This category includes agents designed for specific research tasks, such as those that execute pre-defined algorithms (e.g., TWAP or VWAP) or those that model market impact to test the effects of large trades.
- `Market Makers:` These sophisticated agents contribute to market liquidity by simultaneously quoting bid and ask prices, dynamically adjusting their spreads to manage inventory and risk.

These diverse, rule-based agents form the a robust baseline, allowing us to isolate and study the unique contributions of our LLM-enhanced agents within a complex, realistic ecosystem.

## 3.3 The LLM Agent Layer:

The LLM layer introduces a news-aware reasoning component that influences agent behavior and liquidity. The `EnhancedLLMNewsAnalyzer` accepts structured `NewsEvents` (category, content, affected symbols, sentiment, and importance). It calls the LLM via an `LLMInterface` to analyze the news and normalizes the responses into a quantifiable sentiment and confidence with risk annotations. This output directly informs two key agent types:

- `ABIDESLLMTradingAgent:` This agent translates news analysis into strategy-specific signals (direction, strength, confidence, and duration). It then calculates position deltas using its portfolio value and predefined position caps, placing market or limit orders in the LOB.
- `ABIDESLLMMarketMaker:` This agent quotes around the latest price and dynamically adjusts its strategy. For instance, it widens spreads as uncertainty rises (based on $1 - confidence$ in recent analyses) and re-centers its quotes to manage inventory.

Both agents expose ABIDES-style lifecycle hooks to ensure seamless integration with the simulation kernel and message-passing system. This architecture creates a coupling that is both simple and interpretable: news sentiment drives directional trading pressure, while uncertainty modulates market resiliency and liquidity, producing realistic microstructure responses.

### 3.4 The Brain-Memory Architecture

Our adaptive agent operates on a Brain-Memory Architecture, separating strategic decision-making from persistent, long-term learning.

The "Brain" is the strategic decision-maker, a large, high-capacity model (e.g., Gemini 1.5 Pro). This agent uses a structured prompt to analyze a real-time state vector, including dynamic news headlines and market data, before generating a response.

To overcome context window limitations and enable learning, the "Brain" is supported by a persistent memory module, powered by a file-based database system. Before making a new decision, the "Brain" agent retrieves relevant past experiences from this module. Each memory entry contains the historical news event, the parameters the agent used, the action it took, and the resulting profit or loss. This history is then formatted and injected directly into its prompt. This architecture allows the agent to reflect on its past performance and learn from specific outcomes, a capability absent in traditional agent-based models.

### 3.5 The Multi-Agent Ecosystem: A Three-Layered Approach

Our simulation operates as a multi-layered ecosystem, combining traditional rule-based agents with our novel LLM-powered agents to create a realistic and dynamic market environment.

- `Foundation Layer:` This layer is composed of a diverse population of pre-existing ABIDES agent types, which form the "baseline market." These agents, such as Zero-Intelligence (ZI) agents, Fundamental Value agents, and algorithmic Market Makers, are not intelligent in the human sense but follow predictable, algorithmic behaviors. This population generates a complex, dynamic, and realistic synthetic market microstructure—the liquidity, spreads, and order flow that an agent would see in a real-world exchange.

- `The LLM Integration:` Into this pre-existing, realistic environment, we introduce our LLM-enhanced agents. These agents are not acting in a vacuum or a simplified setting. Instead, they are placed within a competitive and realistic ecosystem populated by dozens of other agents, whose collective actions create a high-fidelity market. The LLM agents' performance and emergent behavior are thus validated against a robust and complex backdrop.

- `The Interaction Loop:` The "Brain" agent's primary function is to interpret the complex, emergent dynamics of the market, which are generated by the diverse population of ABIDES agents. It receives a comprehensive state vector that includes not only high-level information like price and volume but also microstructural details such as order book depth and recent trade prints. The "Brain" then uses its reasoning capabilities to synthesize this information and form a trading hypothesis. The persistent memory module provides a crucial temporal dimension, allowing the "Brain" to ground its real-time decisions in historical context from within the simulated environment. For example, it might recall that a similar news headline in the past led to a specific market reversal, enabling a more nuanced and strategic response.

  Crucially, this feedback loop operates online, within a single simulation run. The outcome of a decision made at hour one is evaluated after a fixed time window and becomes a learned experience—written back to `memory.jsonl`—that informs the agent's strategy at hour two. The orders generated by the "Brain" agent are then fed back into the LOB engine, influencing the market and creating a dynamic feedback loop with the rest of the agent population.

To make the control flow explicit, we summarize the interaction loop in Algorithm 1. At each step $t \in [0, T]$ with step size $\Delta t$, the agent aggregates the market snapshot $m_t$, news items $n_t$, and retrieved memories to form $context_t$, then the Brain proposes a concrete ABIDES configuration $\mathcal{C}_t$ which is validated and executed in the exchange-faithful core. The resulting actions and fills feed metrics that are written back to `Memory` for within-episode adaptation. The next section details how these outputs are aligned to NASDAQ's ITCH protocol for calibration and external validation.

**Algorithm 1** Overall workflow: LLM ↔ ABIDES with Brain–Memory (condensed)

---

**Require:** $H$ (history), $M$ (mkt stream), $N$ (news); Env, $\mathcal{C}_0$; LLM, Memory
**Ensure:** logs, memory snapshot, performance summary
 1: $\mathcal{C} \leftarrow \mathcal{C}_0$; ABIDES $\leftarrow$ Init(Env); Memory.load($H$); set $T, \Delta t$
 2: **for** $t = 0 : \Delta t : T$ **do**
 3: $\quad m_t \leftarrow M.read()$; $\quad n_t \leftarrow N.read()$; $\quad ctx \leftarrow Agg(m_t, n_t, \text{Memory.peek}())$
 4: $\quad$ Memory.update($ctx$); $\quad mem \leftarrow$ Memory.retrieve_relevant_entries($ctx$)
 5: $\quad \mathcal{C}_t \leftarrow$ LLM.generateConfig($\mathcal{C}, mem, ctx$); $\quad \mathcal{C}_t \leftarrow ValidateTranslate(\mathcal{C}_t)$
 6: $\quad$ ABIDES.applyConfig($\mathcal{C}_t$); $\quad (a_t, f_t) \leftarrow$ ABIDES.runStep($\Delta t$)
 7: $\quad met_t \leftarrow Metrics(a_t, f_t, m_t)$; $\quad$ Memory.update_entry_with_outcome($ctx, met_t, f_t$)
 8: $\quad$ **if** learning enabled **then**
 9: $\quad\quad$ LLM.onlineUpdate($met_t, f_t, mem$)
10: $\quad$ **end if**
11: $\quad$ Log($t, \mathcal{C}_t, a_t, met_t, \text{Memory.digest}()$); $\quad \mathcal{C} \leftarrow Carry(\mathcal{C}_t)$
12: **end for**
13: **return** logs, Memory.snapshot(), PerfSummary()

---

### 3.6 Data Ingestion, Calibration, and Validation:

To ensure the realism of our synthetic environment, we ingest and replay real-world financial data at the message level from sources like NASDAQ's ITCH protocol via the LOBSTER dataset. Our system is not merely reproducing historical price curves; it is accurately simulating the underlying market microstructure. The process, governed by scripts like `itch_data_parser.py` and `real_data_ingestion.py`, ensures that our synthetic data (`AAPL_2012-06-21...csv`) maintains the statistical properties of the original. We carefully calibrate the synthetic environment by validating our generated order flow, trade volume, and price volatility against the ground truth of historical NASDAQ data. This calibration relies on an optimization routine (`calibrate_simulator.py`) to find the optimal agent population parameters, stored in `optimal_config.json`, that best reproduce historical market behavior. For external validation, ITCH trades are loaded and aggregated to a per-second baseline. The system then aligns simulated execution timestamps to this per-second clock and compares prices, computing

$$price\_error\_bps = \frac{sim - ITCH}{ITCH} \times 10,000$$

and exporting a per-execution CSV and a metrics JSON. Finally, we plot the smoothed simulated mid-price as a blue line with simulated trades as red scatter over time, using the ITCH first price and start time for a direct visual comparison.

## 4 Results and Discussion

We conducted a series of experiments with our LLM enhanced ABIDES simulator, comparing three conditions under identical seeds, book parameters, market configuration, schedules, and windows, all calibrated to AMZN on June 21, 2012. The three agent populations were LLMON with an adaptive Brain Memory agent, LLMOFF composed of traditional heuristic agents such as simple momentum and contrarian traders, and a Baseline population dominated by noise traders.

In LLMON the news analyzer queries the LLM to generate sentiment, confidence, and risk annotations from structured news events; the trading agent maps these signals to direction and size within a fixed risk budget with position caps and convex scaling; and the market maker widens spreads when confidence is low and recenters quotes based on inventory pressure, yielding news conditioned order flow, adaptive liquidity, and short horizon impact around events. In LLMOFF the analyzer returns deterministic rule based sentiment and confidence without LLM calls, and trading and market making follow the same mappings and risk controls but are driven by heuristic signals, serving as an ablation that isolates the incremental value of LLM reasoning under identical settings. In BASELINE no news is ingested or analyzed, agents run without event driven signals, trading relies on static strategy parameters, and the market maker uses a fixed spread policy, providing a microstructure only baseline that quantifies the contribution of news coupling.

To probe sensitivity to exogenous information, we introduced four synthetic news events via a news-injection interface. As shown in Figure 2, only the LLMON configuration consistently perceived

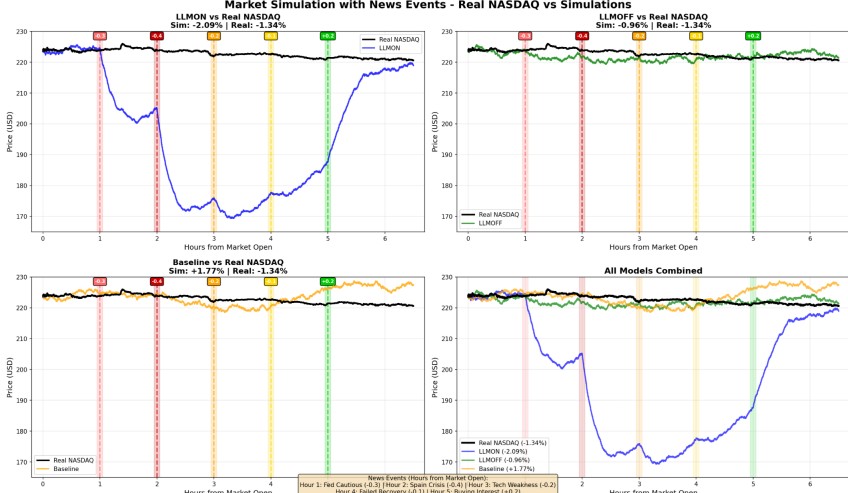

Figure 2: We evaluate the simulator on the LOBSTER dataset using AMZN on June 21, 2012 under three modes: LLM-ON, LLM-OFF, and Baseline. Results with LLM-ON confirm that the simulator can perceive news events and inject their effects into the simulated market.

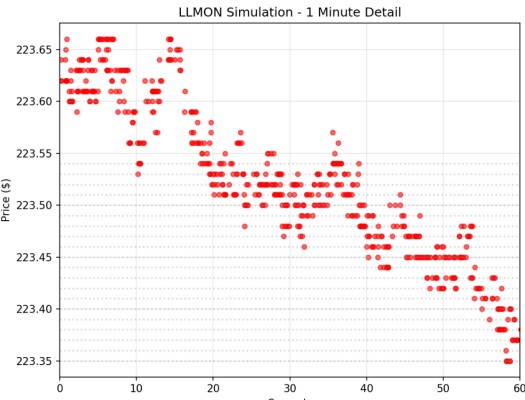

Figure 3: We extract the execution prices of LLM-ON executed trades within a 1-minute interval to illustrate its high-frequency characteristics and rapid temporal dynamics.

these injections and propagated their effects into the simulated price dynamics, whereas LLMOFF and Baseline showed little to no response.

The results demonstrate that the LLMON agent, leveraging its online learning capabilities via the memory module, generates market dynamics that are qualitatively and quantitatively distinct. The introduction of the learning LLM agent was observed to influence the market in ways that traditional agents could not, such as dynamically altering its reaction strength to shifting sentiment from news headlines. The agent's ability to adjust its parameters mid-simulation based on the profitability of its prior actions showcases a strategic evolution absent in the static LLMOFF and Baseline conditions.

Furthermore, as shown in Figure 3, our analysis of the simulated order book and micro structural 1-minute trading data confirms that the adaptive LLM agent contributes to the formation of a more realistic market microstructure. The trade intensity, bid-ask spread, and price volatility in the LLM-enhanced simulation more closely match historical data, showcasing the emergent complexity that arises from an intelligent agent learning within a competitive environment.

Moreover, as shown in Figure 4, our LLMON configuration aligns more closely with the real market environment than either the Baseline or LLMOFF settings. Using identical initial conditions and calibration to AMZN on June 21, 2012, LLMON reproduces key microstructure regularities, including co-movement between spread and volatility, realistic order book depth and imbalance

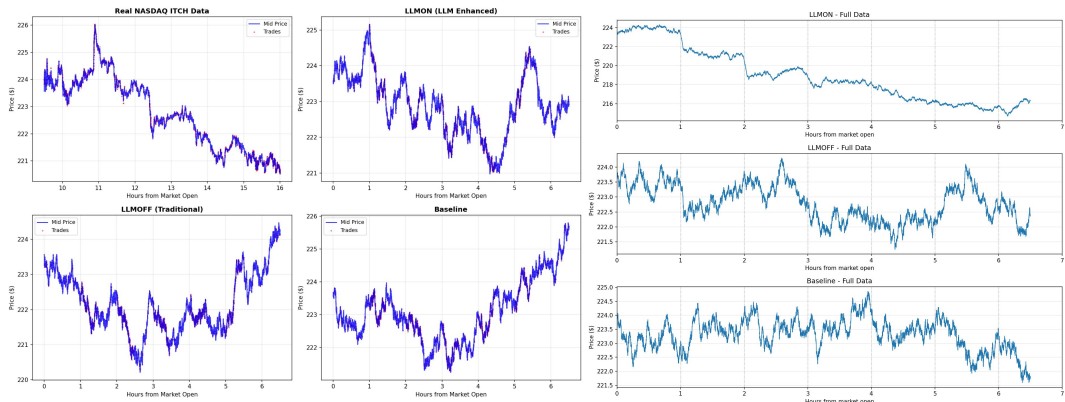

Figure 4: **Left (a):** Real vs. simulated intraday mid-price overlays. **Right (b):** Full-session trajectories for the three simulated configurations. Each panel shows the complete trading day aligned to market open. LLMON exhibits a stronger directional drift, LLMOFF is more range-bound, and Baseline shows a mild decline.

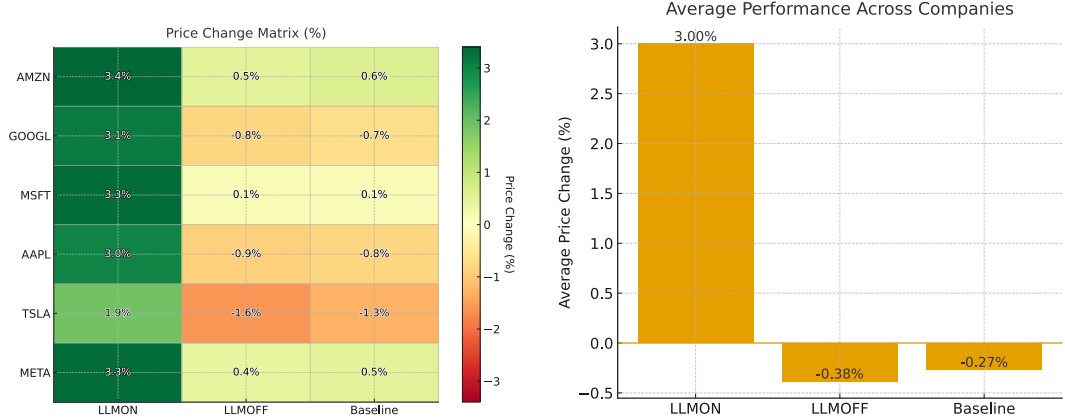

Figure 5: We visualize simulator price responses to exogenous-impact injection using a heatmap. The LLMON setting exhibits a clearer and more accurate reflection of externally driven movements than the alternative modes.

profiles, and a stronger and timely price response to the injected news events, whereas LLMOFF and Baseline show muted or delayed reactions and systematic deviations from the empirical benchmarks.

## 5  Conclusion

We present a high-fidelity, LLM-enhanced financial market simulator that leverages synthetic data and a novel Brain-Memory Architecture to model realistic dynamics. Through an online learning loop, we show that LLM agents can move beyond simple heuristics to exhibit complex, adaptive behaviors, demonstrating the feasibility of using generative AI to simulate emergent phenomena in real markets. Beyond finance, this framework offers a foundation for studying intelligent agents in other high-frequency, adaptive domains such as bonds, emerging markets, and related trading contexts.

Nevertheless, limitations remain: synthetic data cannot fully capture real-world subtleties, and current LLMs face constraints in scale, consistency, and generalization. Addressing these challenges will require integrating heterogeneous data sources, enhancing memory mechanisms, and developing stronger evaluation metrics for emergent behaviors.

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
